# A WRKY Transcription Factor, EjWRKY17, from *Eriobotrya japonica* Enhances Drought Tolerance in Transgenic *Arabidopsis*

**DOI:** 10.3390/ijms22115593

**Published:** 2021-05-25

**Authors:** Dan Wang, Qiyang Chen, Weiwei Chen, Xinya Liu, Yan Xia, Qigao Guo, Danlong Jing, Guolu Liang

**Affiliations:** 1Key Laboratory of Horticulture Science for Southern Mountains Regions of Ministry of Education, College of Horticulture and Landscape Architecture, Southwest University, Beibei, Chongqing 400715, China; wander007@email.swu.edu.cn (D.W.); cqy0909@email.swu.edu.cn (Q.C.); weiwei_chen15134@zzu.edu.cn (W.C.); barophiles@email.swu.edu.cn (X.L.); yansummer@swu.edu.cn (Y.X.); qgguo75@swu.edu.cn (Q.G.); 2Academy of Agricultural Sciences of Southwest University, State Cultivation Base of Crop Stress Biology for Southern Mountainous Land of Southwest University, Beibei, Chongqing 400715, China

**Keywords:** *Eriobotrya japonica*, *WRKY17*, overexpression, drought stress, ABA

## Abstract

The WRKY gene family, which is one of the largest transcription factor (TF) families, plays an important role in numerous aspects of plant growth and development, especially in various stress responses. However, the functional roles of the WRKY gene family in loquat are relatively unknown. In this study, a novel WRKY gene, *EjWRKY17,* was characterized from *Eriobotrya japonica,* which was significantly upregulated in leaves by melatonin treatment during drought stress. The *EjWRKY17* protein, belonging to group II of the WRKY family, was localized in the nucleus. The results indicated that overexpression of *EjWRKY17* increased cotyledon greening and root elongation in transgenic *Arabidopsis* lines under abscisic acid (ABA) treatment. Meanwhile, overexpression of *EjWRKY17* led to enhanced drought tolerance in transgenic lines, which was supported by the lower water loss, limited electrolyte leakage, and lower levels of reactive oxygen species (ROS) and malondialdehyde (MDA). Further investigations showed that overexpression of *EjWRKY17* promoted ABA-mediated stomatal closure and remarkably up-regulated ABA biosynthesis and stress-related gene expression in transgenic lines under drought stress. Overall, our findings reveal that EjWRKY17 possibly acts as a positive regulator in ABA-regulated drought tolerance.

## 1. Introduction

Plants often face multiple environmental constraints in terms of drought stress, temperature extremes, and salinity [1]. Water deficit is the main environmental factor severely constraining plant development and production. To adapt to these abiotic stresses, plants have evolved a series of intricate strategies at molecular, cellular, biochemical, and physiological levels [2,3], including stomatal movement, signal perception and transduction, the expression of stress-induced genes and activation of physiological and metabolic responses [4]. When plants are exposed to stress conditions, the transcription factors (TFs) act as central regulators by binding to specific *cis*-acting elements in the promoter regions to activate downstream gene expression, signal transduction, and adaptation networks [5]. To date, a wide range of TFs families in plants, including WRKY, NAC, MYB, and bHLH, have been functionally identified and elucidated to participate in abiotic stress in plants [6,7,8,9].

As one of the largest transcription factor families, WRKY TFs are crucial regulatory proteins that respond to biotic and abiotic stresses and modulate physiological processes and development [10]. The WRKY family, originally isolated from sweet potato, contain either 1 or 2 WRKY domains, which consist of a conserved WRKYGQK sequence motif and a zinc finger structure (C_2_H_2_ or C_2_HC) [11]. The WRKY gene family is classified into groups I, II, and III and different subsets based on the number of WRKY domains and the variation of zinc finger motif [12,13]. Proteins having two WRKY domains belong to group I, while groups II and III have a single WRKY domain, the latter two groups are distinguished by C_2_H_2_ and C_2_HC, respectively [13]. The modulation of the transcription of downstream target genes can be regulated by WRKY TFs via specifically binding to the W-box elements [(T)TGAC(C/T)] *cis*-elements [12]. Growing evidence has proved that WRKY proteins are involved in plant responses to a broad spectrum of abiotic stresses, such as drought, cold, and salt [6,14,15]. For instance, *AtWRKY21*, *AtWRKY46*, *AtWRKY54*, and *AtWRKY70* in *Arabidopsis* have been reported to regulate osmotic stress [16,17,18]. *TaWRKY2* transgenic wheat enhanced tolerance to drought stress, reduced the water loss rate of detached leaves, and increased grain yield [19]. In addition, overexpression of *OsWRKY11* and *OsWRKY30* effectively enhanced drought tolerance in transgenic rice seedlings [20,21]. *AtWRKY57* increased resistance to drought in *Arabidopsis* through regulation of water loss rate and abscisic acid (ABA) content [22]. Overexpression of *MfWRKY17* in *Arabidopsis* increased water retention, maintained chlorophyll content, and regulated the transcription levels of ABA biosynthesis and the stress-related genes, thus improving the resistance to drought stress [23].

Loquat (*Eriobotrya japonica* Lindl.), belonging to the Rosacea family, is one of the semitropical fruit trees widely distributed in Southeastern China [24,25]. However, the growth, development, and yield of loquat are often affected by multiple abiotic stresses, especially drought and cold stresses [26]. Our previous research showed that the WRKYs family was involved in the drought stress in loquat seedlings [27]. However, the function and molecular mechanism of WRKY TF family members in loquat remains elusive. Here, a novel WRKY gene was cloned from *Eriobotrya japonica* and designated as *EjWRKY17*. The ectopic expression of *EjWRKY17* positively regulates drought tolerance in transgenic *Arabidopsis* through ABA-mediated stomatal closure, inhibition of reactive oxygen species (ROS) accumulation, and up-regulation of the expression of stress-related genes. These results deepen the understanding of the WRKY TFs’ functions in loquat in response to drought stress.

## 2. Results

### 2.1. Identification of EjWRKY17 from Eriobotrya japonica

A WRKY transcription factor gene named *EjWRKY17* was isolated from *Eriobotrya japonica* based on the homology cloning and RACE techniques. The open reading frame of *EjWRKY17* is 837 bp, encoding 278 amino acids (Genebank accession number: MW528209). The calculated molecular weight (MW) was 30.5 kD and the isoelectric point (PI) of the deduced protein was 9.82. As shown in Figure 1A and Appendix A, multiple alignments of amino acid sequences showed that EjWRKY17 protein contained a conserved WRKY domain, consisting of a WRKYGQK motif and a C_2_H_2_ zinc-finger-like motif, which was similar to several WRKY17s with known functions in different plant species. In addition, a total of seven conserved functional domains were identified in the EjWRKY17 protein. Motif 1 was defined as the WRKY domain, and a nuclear localization signal (NLS) (KRRK) was found in Motif 2. Additionally, a conserved C-motif and HARF motif were found in Motifs 4 and 5, respectively. Therefore, EjWRKY17 can be classified into group II of the WRKY family. In Rosacea plants, the protein sequence of EjWRKY17 was most homologous to MdWRKY11 and MdWRKY17 of *Malus domestica* (Figure 1B). EjWRKY17 was phylogenetically closer with VvWRKY17 of *Vitis vinifera* than with AtWRKY17 of *Arabidopsis*, MfWRKY17 of *Myrothamnus Flabellifolia*.

Based on our previous study, exogenous melatonin enhanced drought tolerance in loquat seedlings and significantly induced high expression of *WRKY17* under drought stress at the transcriptome level. To verify the expression pattern of *EjWRKY17* in response to water deficit, loquat seedlings pretreated with or without melatonin were exposed to drought for 13 days (d) and RT-qPCR assays were performed (Figure 1C). Under drought stress, *EjWRKY17* expression in control leaves (CK) significantly decreased from 1 d to 10 d and then increased at 13 d. In contrast, the expression of *EjWRKY17* was induced from 7 d to 13 d in response to exogenous melatonin treatment and the expression levels in melatonin-treated (MT) leaves were significantly higher than that of the control leaves from 7 d to 13 d. These results suggested that *EjWRKY17* gene may be positively involved in the drought resistance of loquat. In addition, we investigated the promoter region of *EjWRKY17* gene for the presence of specific *cis*-acting elements (Figure 1D and Appendix A). The results showed that the *EjWRKY17* promoter contains one MYB binding site (MBS) involved in drought inducibility, one dehydration responsive element (DRE core), one defense and stress-responsive element (TC-rich repeats), and one element related to ABA responsiveness (ABRE), indicating that the EjWRKY17 might be involved in plants’ response to ABA-mediated drought stress.

### 2.2. Subcellular Localization of EjWRKY17 Protein

In order to examine the subcellular location of the EjWRKY17, we constructed *35S::EjWRKY17-GFP* fusion controlled by CaMV 35S promoter. The recombined fusion vector and the control vector (*35S::GFP*) were transferred into the young leaves of tobacco (*Nicotiana benthamiana*) by *Agrobacterium*-mediated transformation. Then the transgenic leaves were stained with 4,6-diamidino-2-phenylindole (DAPI). As shown in Figure 2, the green fluorescence of the control was localized both in the nucleus and cytoplasm, while the signal of the *35S::EjWRKY17-GFP* was only targeted to the nucleus. These findings revealed that the EjWRKY17 is a nuclear-localized protein.

### 2.3. Overexpression of EjWRKY17 Increased ABA Tolerance in Arabidopsis

ABA, as a major chemical signal, plays a crucial role in regulating abiotic stress responses [28]. To observe the response of *EjWRKY17* transgenic plants to ABA treatment, *Arabidopsis* seeds of the T3 generation of WT and *EjWRKY17* overexpression lines (L3 and L4) were surface-sterilized and germinated on half-strength MS medium supplemented with ABA (0, 0.5, and 1 µM). As the concentration increased, ABA treatment inhibited the germination rate of *EjWRKY17* transgenic and WT plants. However, the germination rate was almost identical between the WT and transgenic lines with or without ABA treatment (Figure 3A,B). After six days of growth, there was no significant difference in germination rate between *EjWRKY17* transgenic line and WT plants in the presence of 0.5 or 1.0 µM ABA. Interestingly, the rates of green cotyledon in both *EjWRKY17* transgenic lines were significantly higher than that in the WT at both ABA concentrations (Figure 3C).

In addition, the sensitivity to ABA of *EjWRKY17* overexpression lines was further determined at the post-germination stage. Under normal growth conditions, no phenotypic differences were observed between the WT and transgenic plants. The primary root lengths of *Arabidopsis* were inhibited by ABA stress with the increase of the treatment concentration (Figure 4). In the presence of 1 µM ABA, the root length of *EjWRKY17* transgenic lines was 3.95 cm, about 0.52 cm longer than that of WT plants. Under 5 µM ABA treatment, a more pronounced inhibition of root length was observed in the WT plants compared to the EjWRKY17 overexpression lines. These results suggested that overexpression of *EjWRKY17* enhanced tolerance to ABA at both germination and post-germination stages in *Arabidopsis*.

### 2.4. Overexpression of EjWRKY17 Enhanced Drought Tolerance in Arabidopsis

With the increase of the mannitol concentration, the root growth in WT plants was more severely inhibited than in *EjWRKY17* transgenic plants (Figure 5). When exposed to 150 mM mannitol stress, *EjWRKY17* transgenic lines had notably longer root lengths than the WT line. In the presence of 300 mM mannitol, the root lengths of L3 and L4 lines were constrained to 3.05 and 2.76 cm, significantly higher than that of the WT plants. Taken together, these results indicated that overexpression of *EjWRKY17* in *Arabidopsis* could enhance tolerance to mannitol-induced drought.

To study the function of *EjWRKY17* in response to drought stress, the five-week-old *Arabidopsis* seedlings were exposed to drought stress via water withholding. We observed that most plants wilted with symptoms of leaf yellowing and crimping after two weeks of drought treatment (Figure 6A). WT plants exhibited more severe dehydration and wilted phenotype compared to the *EjWRKY17* transgenic lines. After rewatering for five days, only 32.5 % of WT plants survived, while over 90 % of the *EjWRKY17* transgenic plants returned to normal growth (Figure 6B). Compared to WT plants, a stronger growth was found in the transgenic lines after rewatering. In addition, the water loss rate of detached leaves from WT plants was significantly higher than that of the *EjWRKY17* transgenic leaves at five time points over a 10-h time course (Figure 6C). Electrolyte leakage is an important parameter for evaluating cell membrane damage. Under normal conditions, electrolyte leakage was identical between the transgenic lines and WT plants. However, *EjWRKY17* overexpression in *Arabidopsis* exhibited significantly reduced electrolyte leakage than compared to that of the WT after drought treatment (Figure 6D). These results indicated that the overexpression of *EjWRKY17* improved the drought tolerance in *Arabidopsis* plants.

### 2.5. Overexpression of EjWRKY17 Decreased ROS and Malondialdehyde (MDA) Accumulation under Drought Stress

Drought usually causes lipid peroxidation and the excessive production of ROS, ultimately resulting in oxidative stress [29]. In this study, nitroblue tetrazolium (NBT) and 3,3′-diaminobenzidine (DAB) staining were used to visualize the accumulation of O^2−^ and H_2_O_2_ in *Arabidopsis* leaves and to investigate whether the drought resistance was associated with the rescue of ROS levels in transgenic lines. As shown in Figure 7A,B, no obvious staining in WT or transgenic plants was observed under control conditions. When exposed to drought stress, lower O^2^^−^ and H_2_O_2_ levels were detected in L3 and L4 transgenic lines compared to WT plants. In addition, the MDA content in both transgenic and WT plants dramatically increased when drought stress occurred (Figure 7C). According to our data, the transgenic lines consistently accumulated less MDA than WT plants under drought stress, which suggested that overexpression of *EjWRKY17* regulated ROS homeostasis and reduced the oxidative damage in *Arabidopsis* caused by drought stress.

### 2.6. Overexpression of EjWRKY17 Positively Regulated ABA-Mediated Stomatal Closure in Arabidopsis

We further measured stomatal apertures in *EjWRKY17* transgenic and WT plants under ABA treatment. Under normal conditions, the stomatal aperture indices of L3 and L4 were significantly higher than those of WT (Figure 8). However, in the presence of 1 μM ABA, *EjWRKY17* transgenic lines displayed significantly smaller stomatal apertures than the control. Thus, our results suggested that overexpression of *EjWRKY17* could promote ABA-mediated stomatal closure in *Arabidopsis*.

### 2.7. EjWRKY17 Induced the Expression of Stress-Related Genes under Drought Stress

To elucidate the effects of *EjWRKY17* on the expression of stress-related genes, quantification of the gene expression was conducted. As shown in Figure 9, the expression of *AtABF1*, *AtRD29B*, and *AtLEA14* genes showed no difference under normal conditions, while transcript levels of these three genes were significantly increased in transgenic lines and WT plants under drought stress. Compared with WT plants, overexpression of *EjWRKY17* notably enhanced the transcripts of these genes. In addition, four stress-related genes (*AtRD29A*, *AtCOR15A*, *AtRAB18*, and *AtKIN1*) in transgenic lines were prominently up-regulated when seedlings were under normal conditions or exposed to drought conditions. The expression levels of *AtRD22* and *AtLEA76* in the *EjWRKY17*-overexpressing lines were significantly lower than in WT plants under normal conditions. However, the expression levels of *AtRD22* and *AtLEA76* were greatly upregulated in the *EjWRKY17*-overexpressing lines under drought stress, which were significantly higher than in WT plants. To further study how EjWRKY17 regulates these downstream genes, we analyzed the promoter regions of these marker genes with the PlantCARE database (Appendix A). One, three, one, three, five, two, and four W-box *cis*-elements were found in the up-stream sequences of *ABF1*, *RD29A*, *RD29B*, *RD22*, *LEA14*, *LEA76,* and *KIN1*, respectively. In addition, two WK box (TTTTCCAC) *cis*-elements were found in the up-stream sequences of *RAB18*. Therefore, we speculated that EjWRKY17 may directly or indirectly regulate expression of drought-related genes by binding to the W/WK-box in their promoter regions, thereby responding to drought stress and ABA signaling.

## 3. Discussion

Drought directly affects the growth and productivity of plants. Under drought stress, the total chlorophyll content and photosynthetic rates of loquat leaves were decreased, resulting in a decline in starch content [27]. Severe drought reduced all growth parameters and particularly leaf growth [26]. Our previous research demonstrated that the WRKY family is involved in the enhancement of drought tolerance in loquat seedlings at the transcriptome level. WRKY TFs family are known to mediate abiotic stress responses in various plants; the information of its functions mainly focuses on *Arabidopsis* [16], rice [20], and soybean [30]. However, the functions and mechanisms of the WRKY gene family in loquat are still unclear. In this study, *EjWRKY17*, a member of the loquat WRKY family, was characterized to explore its potential role in drought tolerance. The multiple alignments of amino acid sequence and phylogenetic tree analysis of EjWRKY17 exhibited that the EjWRKY17 protein classified into subgroup II shared high similarity with MdWRKY11, MdWRKY17, and other WRKY proteins (PaWRKY11, JrWRKY17, PtWRKY51, and GsWRKY17). *Cis*-acting elements play an important role in gene transcription and expression [31]. The results showed that many types of *cis*-acting elements were exhibited in the EjWRKY17 promoter region, including hormone response elements (AAGAA-motif, ABRE, AuxRR-core, CGTCA-motif, ERE, TGACG-motif, CGTCA-motif, MYB-like sequence, and P-box), stress response elements (STRE, TC-rich repeats, 3-AF1, ARE, AE-box, Box 4, LTR, TCCC-motif, WRE3, and as-1), W-box, and plant physiological metabolism-related elements (O2-site, MYB recognition site, circadian, MBS, and GCN4-motif), suggesting that the EjWRKY17 protein was involved in stress and hormone response [32]. The subcellular localization assay displayed that the EjWRKY17 protein was localized to the cell nucleus. A previous study reported that WRKY17 was involved in conferring stress tolerance in *Arabidopsis* [33]. Additionally, The WRKY transcription factors function as positive or negative regulators in salt and drought stress response in different species [23,34]. In the present study, overexpression of *EjWRKY17* in *Arabidopsis* enhanced green cotyledons formation and root elongation under ABA treatment. Besides, the root growth inhibition was alleviated in the transgenic lines under mannitol stresses compared with WT plants. Moreover, the adult transgenic lines showed a distinct resistance to water deficit with a higher survival rate in comparison with WT plants. These results were also confirmed in other plants, such as *FtWRKY46* in *Tartary buckwheat* [6] and *HbWRKY82* in *Hevea brasiliensis*. In addition, overexpression of *ZmWRKY17* resulted in decreased ABA sensitivity through regulating the expression of several well-known stress/ABA-responsive genes in *Arabidopsis* [35]. Our results suggested that ectopic expression of *EjWRKY17* resulted in decreased sensitivity to ABA and osmotic stress in *Arabidopsis*.

The excessive ROS production triggered by drought caused lipid peroxidation, protein oxidation, nucleic acid damage, and even cell death in plants [36]. When loquat seedlings were subjected to drought stress, the MDA content of loquat leaves exhibited an increase followed by a gradual drop [27]. The MDA concentration is a parameter of membrane lipid peroxidation, which reflects the extent of stress [37]. In this study, transgenic plants suffered lower degrees of membrane injury under drought stress in comparison with WT plants, as indicated by reduced accumulation of electrolyte leakage, ROS, and MDA. Hence, these results revealed that the enhanced capacity of the ROS-scavenging system is highly associated with increased drought tolerance in *EjWRKY17*-overexpressing plants. Our results are consistent with the previous study that *IbWRKY2* confers drought resistance in *Arabidopsis* through activating the ROS-scavenging system [5].

The response pathways to drought mediated by TFs were described as ABA dependent and ABA independent [38]. In the WRKY family, many genes have been reported to participate in the ABA-mediated stress signaling pathway [38]. For instance, *WRKY18* and *WRKY60* conferred hypersensitivity to ABA stress in inhibiting seed germination and root growth, while WRKY40 antagonized WRKY18 and WRKY60 in response to sensitivity to ABA and the abiotic stress [39]. The ABA content of loquat leaves gradually accumulated and ABA biosynthetic genes were significantly regulated under drought stress [27]. Moreover, ABA-mediated stomatal closure provided a vital strategy in plant stress responses to prevent water loss [40]. In recent years, several WRKY TFs have been characterized as regulating stomatal closure. The overexpression of *GmWRKY16* in *Arabidopsis* resulted in a water loss rate and reduction of stomatal aperture under drought stress [29]. In a previous study, *MfWRKY17* from *Myrothamnus flabellifolia* was confirmed as a positive regulator to mediate stomatal closure through an ABA-dependent signaling pathway [23]. Similarly, *EjWRKY17* overexpression in *Arabidopsis* led to enhanced tolerance to ABA stress and promoted ABA-induced stomatal closure. Correspondingly, a decrease in the water loss rate was observed in transgenic detached leaves. The results indicate that *EjWRKY17* has positive regulatory functions in the drought tolerance of plants through an ABA-dependent signaling pathway.

Extensive research has exhibited that WRKYs TF family positively regulated tolerance to drought stress through up/down-regulating the levels of stress/ABA-responsive genes [13,41]. For instance, overexpression of *TaWRKY19* conferred tolerance to salt and drought through up-regulating the expression of *DREB2A*, *RD29A*, and *RD29B* in transgenic *Arabidopsis* [42]. *RD29A*, considered to be a marker gene of the ABA-dependent pathway, is related to reducing stress injury in plants [6]. *WRKY57* can directly bind the W-box of *RD29A* promoter sequences to enhance drought tolerance in *Arabidopsis* [22]. Overexpression of *GmWRKY16* significantly up-regulated the expressions of stress-related marker genes such as *RD29A*, *RD22*, *KIN1*, and *LEA14* in transgenic plants under drought stress. Conversely, the negative regulation of WRKY17s in other plant species has been reported. For instance, the expression levels of *AtRD29A* and *AtDREB2B* were notably down-regulated in *CmWRKY17* transgenic *Arabidopsis* under salinity stress [43]. In contrast, the expression of stress-responsive genes (*ABF1*, *RD29A*, *RD29B*, *RAB18*, *RD22*, and *COR15A*) and stress-related marker genes (*LEA14*, *LEA76*, and *KIN1*) occurred at much higher levels in these three transgenic lines than in WT plants under drought stresses in this study. In addition, overexpression *of HaWRKY76* enhanced drought tolerance of transgenic *Arabidopsis* lines by regulating the expression of *RAB18*, *RD29A*, and *RD29B* [44]. Moreover, GhWRKY59 directly binds to the W-boxes of *GhDREB2*, which encodes a dehydration-inducible transcription factor regulating ABA-independent drought response [45]. WRKY transcription factors were reported to regulate the expression of target genes by binding to W-box or WK box in their promoter regions [46]. Hence, we analyzed the promoter sequences of *ABF1*, *RD29A*, *RD29B*, *RAB18*, *RD22*, *LEA14*, *LEA76*, *KIN1,* and *COR15A* with the PlantCARE analysis database, and observed the binding sites existed in the up-stream sequences of these genes except *COR15A*. We assumed that *EjWRKY17* may respond to drought stress by directly or indirectly regulating the drought stress-related genes.

## 4. Materials and Methods

### 4.1. Plant Materials

Grafted *Eriobotrya japonica* ‘Huabai No.1′ was potted at the experimental farm of Southwest University, Chongqing, China, and the seedlings were irrigated with water (CK) or melatonin (MT) for 15 d and then subjected to drought stress by withholding their watering for 13 d, at which point the plants displayed phenotypic foliar wilting [27]. The leaves were collected at 1, 4, 7, 10, and 13 d and immediately frozen in liquid nitrogen. *Nicotiana benthamiana* was used for transient expression. Wild type (*A. thaliana* ecotype) and transgenic plants (homozygous T3 generation) were cultured in a humidity-controlled environment (16 h light/8 h dark cycles, 20 ± 1 °C).

### 4.2. Cloning and Sequence Analysis of EjWRKY17

Total RNA extraction, first-strand cDNA synthesis, and qPCR were performed as described previously [47]. The 5′ and 3′ RACE clones were amplified with the SMARTer RACE 5′/3′ Kit (Clontech, Mountain View, CA, USA) and 3′-Full RACE Core Set (TAKARA, Shiga, Japan). The amplicon of partial length was obtained based on the homology cloning, and then the gene was made full length by 5′ and 3′ RACE. The sequences were joined to obtain an open reading frame of EjWRKY17 cDNA. The specific primers are listed in Appendix A.

The multiple sequence alignment was performed by ClustalX using BioEdit v7.1.11 software (Ibis Biosciences, Carlsbad, CA, USA), and the phylogenetic analysis was constructed with the MEGA software (version 5.0, Auckland, New Zealand) under the Nerghbor-Joining method with 1000 bootstrap replicates [32] with other sequences (Appendix A) obtained from NCBI (https://www.ncbi.nlm.nih.gov/) (accessed on 30 April 2021). MEME (https://meme-suite.org/meme/) (accessed on 30 April 2021) was employed for the analysis of conserved motifs in the WRKY protein. The 2100 bp promoter DNA sequence upstream of the ATG start codon was extracted from the genome (accession No. PRJNA579885) in NCBI. *Cis*-acting elements in the promoter region were predicted and analyzed using PLANTCARE (http://bioinformatics.psb.ugent.be/webtools/plantcare/html/) (accessed on 10 May 2021).

### 4.3. Subcellular Location of EjWRKY17 Protein

The coding sequence (without stop codon) of *EjWRKY17* was cloned into a pCAMBIA 1300-GFP vector and the specific primers with restriction sites (*BamH*I and *Xba*I) are listed (Appendix A). Then, the constructed EjWRKY17-GFP fusion protein and the empty vector were transformed into *Agrobacterium tumefaciens* strain (GV3101) and transiently expressed in the leaves of *Nicotiana benthamiana*. The fluorescence signals were captured with a fluorescence microscope Observer DP80 (Olympus, Tokyo, Japan).

### 4.4. Arabidopsis Transformation

The coding sequence of *EjWRKY17* was cloned into the overexpression vector pFGC5941 with CaMV 35S promoters to generate the recombinant vector. Then, the resultant construct was introduced into *Agrobacterium tumefaciens* strain (GV3101) and transformed into *Arabidopsis* Col-0 by the floral dip method [48]. The transgenic lines were screened by 20 mg/L glufosinate-ammoniums and confirmed by PCR analysis (Appendix A and Appendix A). Two T3 homozygous lines (L3 and L4) with relatively higher expression were selected for observing phenotype and further analysis.

### 4.5. Seedling Growth Assays

For seed germination assays, seeds of homozygous T3 transgenic lines and WT were sown on half-strength MS agar media with ABA (0, 0.5, and 1.0 µM). Seeds were stratified at 4 °C for 2 days under darkness and then moved to a climate-controlled room at 16-h light. 8-h dark cycle and 23 °C for four days.

For root elongation assays, *Arabidopsis* seedlings were grown normally on half-strength MS media for four days. Seedlings with similar growth status were then transferred to half-strength media containing ABA (0, 1.0, and 5.0 µM) and mannitol (0, 150, and 300 mM), respectively. The length of the primary root was recorded after 4 days with three replications.

### 4.6. Drought Stress Tolerance Assays

For drought tolerance assays, four-week-old soil-grown transgenic *EjWRKY17* lines and WT plants under normal conditions were stopped irrigating until stress symptoms occurred, and the seedlings were then re-watered for recovery. For the determination of electrolyte leakage, sample leaves (0.1 g) were rinsed in 10 mL of distilled water, and then the electrolyte leakage in the solution (E1) was measured after 22 h of floating at room temperature by conductivity meter (DDS-309^+^, Chengdu, China). Subsequently, the leaves were boiled for 20 min. After the solution had cooled to room temperature, the conductivity of the solution (E2) was recorded. The relative electrolyte leakage was evaluated as (E1/E2) × 100 % [49]. Water loss data were recorded for ten detached leaves of WT and transgenic *Arabidopsis* lines at 0, 0.5, 1, 2, 4, 6, 8, and 12 h.

### 4.7. Measurement of the Content of ROS and MDA

DAB and NBT staining were performed to detect the accumulation of H_2_O_2_ and O^2−^ as previously described [50]. Meanwhile, the MDA content was determined according to the previous study [51].

### 4.8. Stomatal Aperture Analysis

For stomatal aperture assay, rosette leaves of four-week-old *Arabidopsis* seedlings were collected and incubated in 200 mL stomatal closure solution (10 mM KCl, 1 mM CaCl_2_, 25 mM MES) for 2 h [52]. Photographs were taken using an optical microscope (Hitachi, Tokyo, Japan). The width to length ratio of stomatal was calculated using ImageJ software (version 1.8.0, National Institutes of Health; Bethesda, MD, USA).

### 4.9. Quantitative Real-Time Polymerase Chain Reaction (qRT-PCR)

qRT-PCR was conducted to measure changes in stress-related gene expression by using qTOWER^3^ G (Analytik Jena, Jena, Germany). PCR amplification procedures was set as following: 95 °C for 30 s, followed by 40 cycles at 95 °C for 20 s, 56 °C for 60 s, and a melt cycle from 65 to 95 °C. All primer sequences are listed in Appendix A and the loquat *qEjActin* gene and *Arabidopsis AtActin* gene were used as internal controls. PCRs were performed in 10-μL volumes using NovoStart ^®^SYBR qPCR SuperMix Plus (Novoprotein, Shanghai, China) with each primer at a concentration of 0.05 μM. The 2^−∆∆Ct^ method [53] was used to calculated the relative expression values.

### 4.10. Statistical Analysis

Each experiment was based on three independent biological replicates of each sample and three technical replicates of each biological replicate. The data were shown as the means ± SD. Significant analysis was evaluated by Student’s t-test and SPSS 19.0 software (SPSS Inc., Chicago, IL, USA) with One-way ANOVAs analysis.

## 5. Conclusions

In summary, our results suggest that EjWRKY17 is a drought-related TF located in the nucleus. Overexpression of *EjWRKY17* decreased the ABA sensitivity level of transgenic *Arabidopsis*. Moreover, the inhibition of water loss rate, electrolyte leakage, ROS, and MDA accumulation further confirmed the enhanced drought tolerance in the *EjWRKY17* transgenic lines. Furthermore, we also proved that *EjWRKY17* improved tolerance to drought in transgenic *Arabidopsis* by promoting ABA-induced stomatal closure and activating stress-related genes. Our data provide insights into the regulation of drought tolerance by *EjWRKY17,* while the function of *EjWRKY17* in response to other stresses needs to be further investigated.

## Figures and Tables

**Figure 1 ijms-22-05593-f001:**
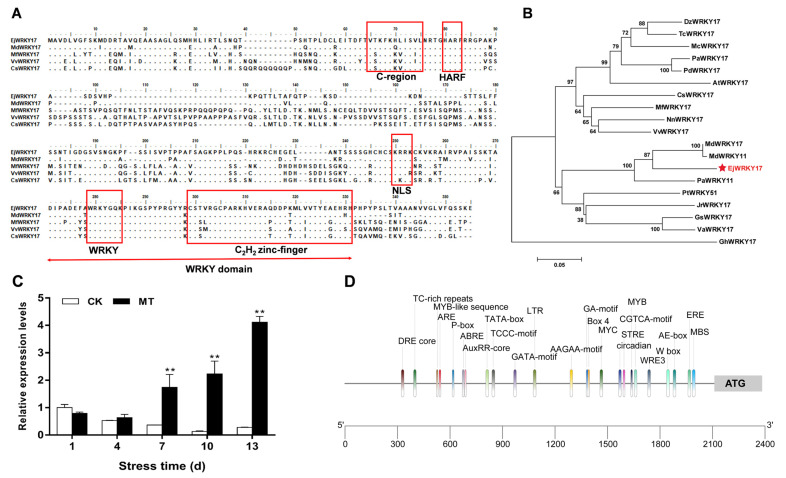
Characterization of EjWRKY17. (**A**) Multiple alignment of amino acid sequences of EjWRKY17 with other WRKY proteins. (**B**) Phylogenetic analysis among EjWRKY17 and WRKY proteins from other plant species. (**C**) Expression levels of *EjWRKY17* during drought stress. (**D**) *Cis*-acting elements in the promoter region of *EjWRKY17* gene. The conserved domain or motif was marked by the red line; NLS, nuclear localization signal. Asterisks denote significant differences (as compared with the control group): ** *p* < 0.01.

**Figure 2 ijms-22-05593-f002:**
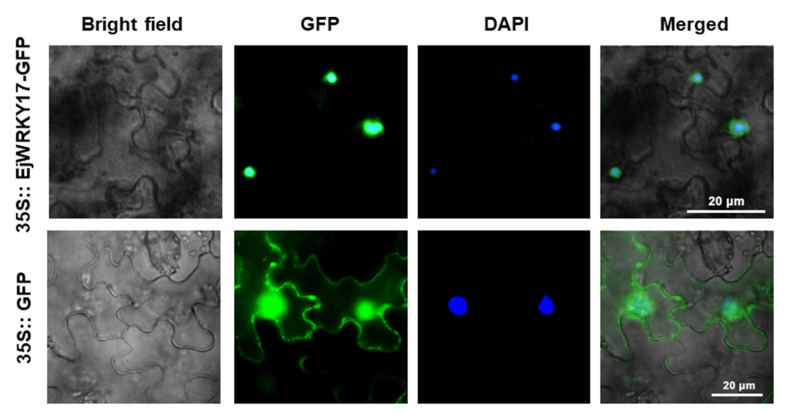
The subcellular localization of EjWRKY17 in *N*. *benthamiana* epidermal cells. GFP: Green fluorescence protein. DAPI: Nucleic marker. Merged: Merged image of bright field, GFP, and DAPI.

**Figure 3 ijms-22-05593-f003:**
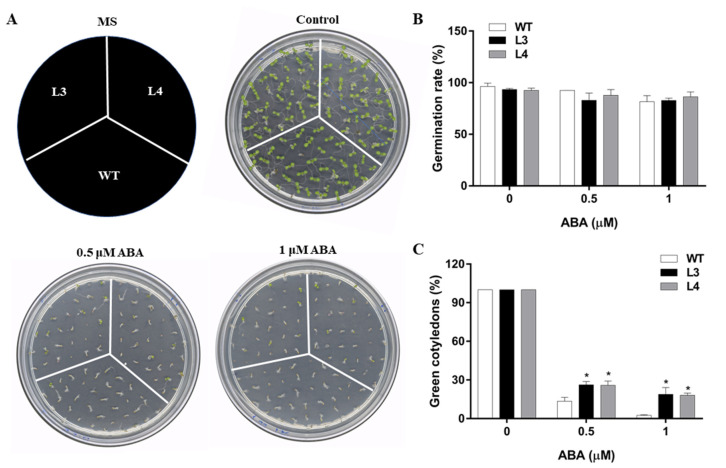
Germination and green cotyledon analysis of *Arabidopsis* under ABA treatment. (**A**,**B**) Germination rates. (**C**) Cotyledon greening rates. Error bars represent ± SD (*n* = 3). Asterisks denote significant differences (as compared with the control group): * *p* < 0.05.

**Figure 4 ijms-22-05593-f004:**
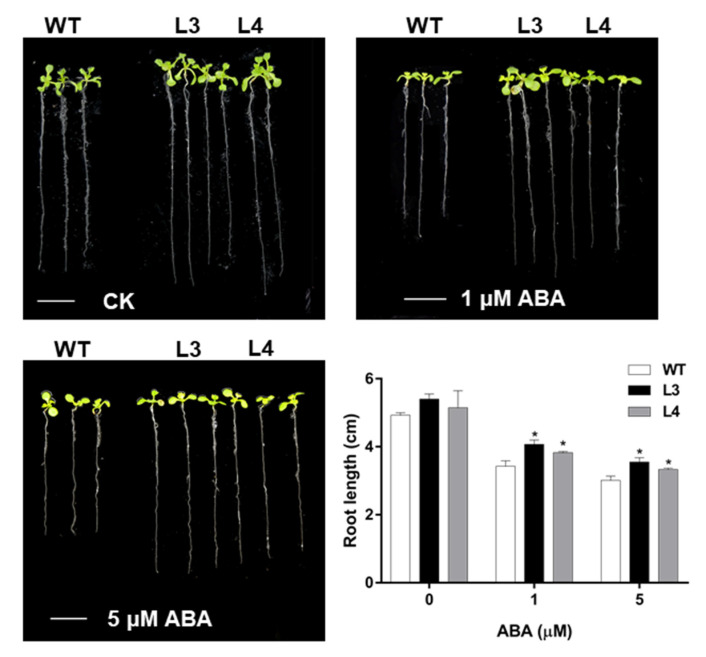
Root elongation analysis of *Arabidopsis* under ABA treatment. Error bars represent ± SD (*n* = 3). Asterisks denote significant differences (as compared with the control group): * *p* < 0.05.

**Figure 5 ijms-22-05593-f005:**
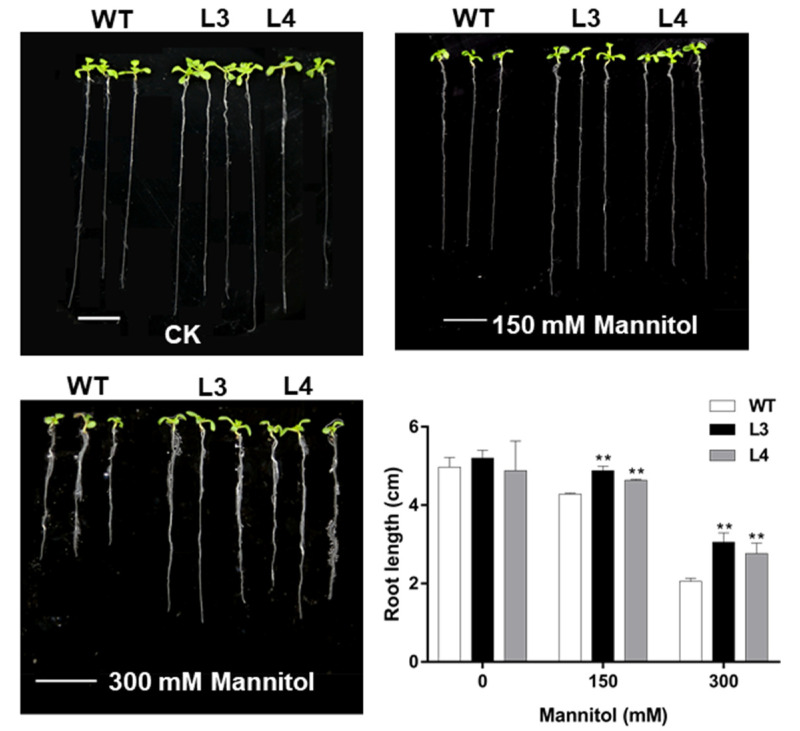
Root elongation analysis of *Arabidopsis* under mannitol treatment. Error bars represent ± SD (*n* = 3). Asterisks denote significant differences (as compared with the control group): ** *p* < 0.01.

**Figure 6 ijms-22-05593-f006:**
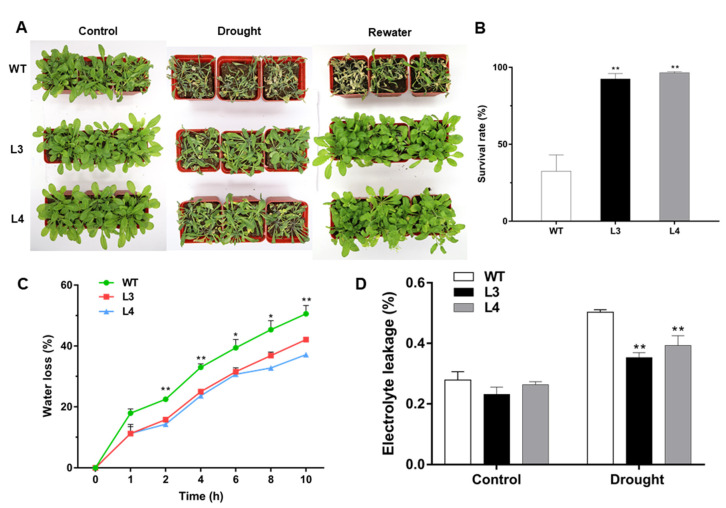
Overexpression of *EjWRKY17* confers tolerance to drought stress in *Arabidopsis*. (**A**) Phenotype of *Arabidopsis* with overexpression of *EjWRKY17* under water deficit. (**B**) Survival rates of seedlings. (**C**) The water loss of detached leaves. (**D**) Electrolyte leakage. Plants were photographed on 2 d (control) and 15 d (drought) without irrigation and on 3 d after rewatering (rewater). Error bars represent ± SD (*n* = 3). Asterisks denote significant differences (as compared with the control group): * *p* < 0.05; ** *p* < 0.01.

**Figure 7 ijms-22-05593-f007:**
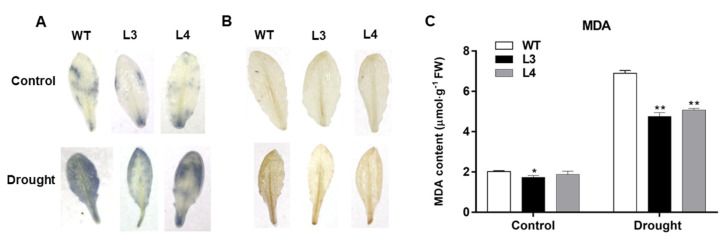
Overexpression of *EjWRKY17* decreased ROS production and oxidative damage under drought stress. (**A**) NBT staining, (**B**) DAB staining, (**C**) MDA content. Error bars represent ± SD (*n* = 3). Asterisks denote significant differences (as compared with the control group): * *p* < 0.05; ** *p* < 0.01.

**Figure 8 ijms-22-05593-f008:**
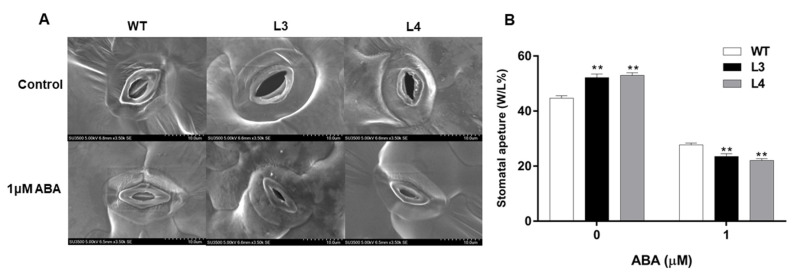
The stomatal aperture of WT and *EjWRKY17* transgenic lines under ABA treatment (**A**,**B**). Error bars represent ± SD (*n* = 3). Asterisks denote significant differences (as compared with the control group): ** *p* < 0.01.

**Figure 9 ijms-22-05593-f009:**
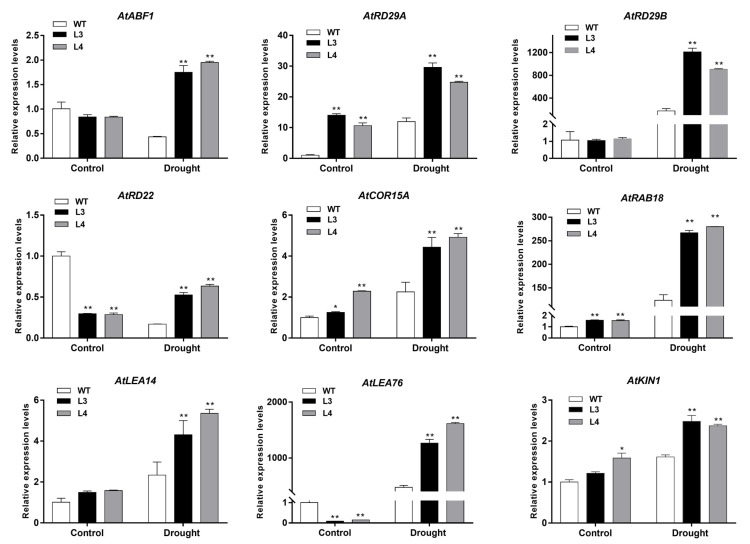
Expression levels of stress-responsive genes in WT and transgenic lines under drought stress. Error bars represent ± SD (*n* = 3). Asterisks denote significant differences (as compared with the control group): * *p* < 0.05; ** *p* < 0.01.

## Data Availability

EjWRKY17 sequence data is available in GenBank.

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
