# Peer review of "A WRKY Transcription Factor, EjWRKY17, from Eriobotrya japonica Enhances Drought Tolerance in Transgenic Arabidopsis"

_ijms, 2021, doi:10.3390/ijms22115593_

Round 1
Reviewer 1 Report
The paper by Wang et al. describes the generation and characterization of an Arabidopsis line overexpressing WRKY17 from loquat. The found that overexpression of this TF leads to enhanced drought tolerance. While overexpression of Arabidopsis WRKY17 was already shown to enhance drought tolerance, there is some difference between the At and Ej protein sequence. This research suggests the phenotype provided by WRKY17 OE is robust.
The manuscript is scientifically sound but could improve from another round of correction for the English language, there are some strange sentences throughout the text.
Further the authors should clearly indicate if the three replicates are biological or technical replicates.
Also indicate in the M&M section which reference genes were used for the qPCR analysis.
Please provide info on the Agrobacterium strain used for the infiltration
Line 23, ‘C2H2’ please make uniform with subscript (as in line 52 etc)
For the graphs in the figures, statistical differences with the control were indicated. Could the authors also indicate if there are statistical differences between the two lines?
Suggestion, ‘half MS’ change to ‘half-strength MS’
Reviewer 2 Report
The focus of the ms. by Wang et al. is on the functional role WRKY TFs in loquat. Thus, EjWRKY17 was cloned and functionally characterized in loquat. Ectopic expression of EjWRKY17 resulted in positive regulation of drought tolerance in transgenic Arabidopsis through ABA-mediated stomatal closure, inhibition of ROS accumulation, and up-regulation of the expression of stress-related genes. Although the experiments are well carried out, and the results may be important for understanding the functional roles of WRKY TFs in loquat, the results are not really unexpected based on studies of other plants and WRKY TFs, and therefore of limited interest
- From reading the ms. it is not clear what constitutes a WRKY domain. Is it just the short WRKY motif, or does the domain also include zinc fingers.
- Some of the figure legends are too short, especially those of Fig. 1 and 2, which simply do not explain the figures.
- The study examines the effects of EjWRKY17 on expression of generic stress-responsive genes. However, the work would benefit from the identification of EjWRKY17-specific genes, either experimentally or at least based on prediction of WRKY promoter binding sites. This would provide loquat-specific information, which is currently actually limited in the ms, remembering that ectopic expression was performed in Arabidopsis.
- Furthermore, the Discussion is not focused on loquat, but on WRKY-function in general. It should be better explained how this improves knowledge in relation to loquat.
Round 2
Reviewer 2 Report
There was a misunderstading in the response to my comment
"Point 3: The study examines the effects of EjWRKY17 on expression of generic stressresponsive genes. However, the work would benefit from the identification of EjWRKY17-specific genes, either experimentally or at least based on prediction of WRKY promoter binding sites. This would provide loquat-specific information, which is currently actually limited in the ms, remembering that ectopic expression was performed in Arabidopsis.
Response". It referred to the prediction and determination of genes regulated by EjWRKY, not the other way around.
